# Alcohol, tobacco and drug use among adults experiencing homelessness in Accra, Ghana: A cross-sectional study of risk levels and associated factors

**Benedict Osei Asibey**[1◉]*, **Brahmaputra Marjadi**[1,2◉], **Elizabeth Conroy**[1◉]

1 Translational Health Research Institute, Western Sydney University, Sydney, New South Wales, Australia,
2 School of Medicine, Western Sydney University, Sydney, New South Wales, Australia

◉ These authors contributed equally to this work.
* b.oseisibey@westernsydney.edu.au

## Abstract

### Background

Substance use contributes to poor health and increases the risk of mortality in the homeless population. This study assessed the prevalence and risk levels of substance use and associated factors among adults experiencing homelessness in Accra, Ghana.

### Methods

305 adults currently experiencing sheltered and unsheltered homelessness in Accra aged ≥ 18 years were recruited. The World Health Organization's (WHO) Alcohol, Smoking, and Substance Involvement Screening Test (ASSIST) was used to assess substance use risk levels. Association of high-risk substance use with sociodemographic, migration, homelessness, and health characteristics were assessed using logistic regression.

### Results

Nearly three-quarters (71%, n = 216) of the sample had ever used a substance, almost all of whom engaged in ASSIST-defined moderate-risk (55%) or high-risk (40%) use. Survivors of physical or emotional violence (AOR = 3.54; 95% confidence interval [CI] 1.89–6.65, p<.001) and sexual violence (AOR = 3.94; 95%CI 1.85–8.39, p<.001) had significantly higher odds of engaging in high-risk substance use, particularly alcohol, cocaine, and cannabis. The likelihood of engaging in high-risk substance use was higher for men than women (AOR = 4.09; 95%CI 2.06–8.12, p<.001) but lower for those in the middle-income group compared to low-income (AOR = 3.94; 95%CI 1.85–8.39, p<.001).

### Conclusions

Risky substance use was common among adults experiencing homelessness in Accra, and strongly associated with violent victimisation, gender, and income levels. The findings highlight the urgent need for effective and targeted preventive and health-risk reduction

**Funding:** The author(s) received no specific funding for this work.

**Competing interests:** The authors have declared that no competing interests exist.

**Abbreviations:** ASSIST, Alcohol, Smoking, and Substance Involvement Screening Test; AUDIT, Alcohol Use Disorder Identification Test; CIDI, Composite International Diagnostic Interview; DAST, Drug Abuse Screening Test; ETHOS, European Typology of Homelessness and Housing Exclusion; GLSS, Ghana Living Standard Survey; GSS, Ghana Statistical Service; ISSER, Institute of Statistical, Social and Economic Research; SSI, Specific Substance Involvement; UNODC, United Nations Office on Drugs and Crime; US, United States; WHO, World Health Organisation.

strategies to address risky substance use in the homeless population in Accra and similar cities within Ghana and sub-Sahara Africa with a high burden of homelessness.

## Introduction

Homelessness affects a growing number of people worldwide, and the homeless population is susceptible to various health problems [1–3]. A critical issue regarding the health of people with experience of homelessness is substance use [4–7]. Among people experiencing homelessness, problematic substance use contributes to a high proportion of poor health and increased risk of mortality through overdoses, infectious diseases, and psychiatric conditions [8, 9]. For example, tobacco use contributes to respiratory conditions, cancers, and heart disease [10, 11], and risky use of alcohol is associated with digestive problems, ulcers, high blood pressure, anxiety and depression, and liver diseases [12, 13].

In Ghana, both homelessness [14, 15] and substance use [16–18] constitute significant public health problems. Urban homelessness has been increasing in Ghana for decades [14, 15]. There is no census data on homelessness, but estimates have varied between 100,000 to 120,000 [19, 20]. Homelessness prevalence is likely to be substantially underestimated in Ghana due to a narrow definition that only includes rooflessness or rough sleeping [21–23]. An unknown proportion of the estimated 39% of Ghana's urban population (5.5 million people) living in slums [24] may be counted as homeless under a broader definition. Various studies in Ghana have revealed a high prevalence of alcohol and drug use in the general population. Ghana is among the leading consumers of illicit drugs: first in Africa and third in the world for marijuana, and first in Africa and 14th in the world for cocaine [25]. Alcohol, tobacco, methamphetamine, opioids, and other illicit drug categories are also highly consumed in Ghana [17, 26].

Despite the high prevalence of homelessness and substance use in Ghana, research on problematic substance use in the homeless population is scant. In fact, the only relevant study found was limited to street children aged up to 16 years [16]. Insufficient knowledge of substance use among the adult homeless population in Ghana is concerning since a large proportion of the homeless population in Ghanaian cities are adult urban migrants (both internal and external to Ghana) [23]. In addition, substance use in the Ghanaian general population is highly prevalent among adults aged 18 years or older in urban areas [17, 26]. Therefore, this study describes the spectrum of lifetime and recent substance use among adults with experience of homelessness in Ghana's capital city, Accra. The study also reports the risk levels and examines factors associated with high-risk substance use. These analyses are important in helping policymakers and service providers tailor policies, treatment, and services to improve the health and well-being of people experiencing homelessness.

## Methods

### Study design and setting

The study was part of a larger cross-sectional community and facility-based research to explore homelessness as a public health phenomenon in Accra, Ghana and investigate the health and health service utilisation of people experiencing homelessness. This paper was based on the second of five empirical studies that included pathways to homelessness, substance use, health impacts of homelessness, social stigma and health care utilisation, and homelessness survival. Accra is Ghana's capital, the most populous city, and the most popular destination for both internal and international migrants [23, 27]. The city has the largest housing deficit in Ghana

and a high prevalence of homelessness [23]. The study was conducted in areas highly populated with people who sleep rough, squatters, and in crisis/emergency shelters and residential drug and alcohol rehabilitation facilities.

## Study population

The study adopted the inclusive European Typology of Homelessness and Housing Exclusion (ETHOS) definition [28], where a person was considered as experiencing homelessness if they met at least one of the following conditions:

1. Unsheltered homelessness:

   a. Sleep rough on the streets, marketplaces, abandoned vehicles and buildings, under bridges, verandas, and unauthorised places not designed for shelter.

   b. Reside in temporal/ non-conventional structures such as makeshift dwellings (slums, shacks, and squats) in extreme overcrowding and mostly under threat of eviction.

2. Sheltered homelessness:

   a. Temporarily reside in a homeless shelter/crisis/emergency accommodation.

   b. Women living in a shelter because of the experience of domestic violence and where the period of stay is intended to be short-term.

   c. Has been admitted to a residential drug rehabilitation with an immediately preceding homelessness and no accommodation when discharged.

Due to the overwhelming predominance of the extended family system in the Ghanaian culture [29–31] where close and remote relatives commonly reside in compound housing style [32], living in conventional housing temporarily with relatives was not categorised as homelessness in this research. The 2000 and 2010 population and housing census reports showed that compound houses generally referred to as 'family houses' constituted 55% of the housing stock in urban Ghana [23].

A sample size of 200 was initially planned but 325 eligible individuals consented, out of which 305 (94%) completed the survey. Twenty individuals did not complete due to conflicting work and counselling appointments. Since there was no official data on homelessness in Ghana, the uncontrolled quota sampling technique [33, 34] was used to obtain a representation of sheltered and unsheltered homeless populations. This technique was adopted because it allows the researchers the freedom to choose sample group members according to their will and/or knowledge without any restrictions. Based on the lead researcher [BOA]'s knowledge and experience in working with the homeless population in Accra, and prior reports of the unsheltered homeless being the prevalent group in Ghana [14, 20], a greater number of eligible unsheltered homeless was sought as compared to the sheltered. The initial plan was to sample 70% unsheltered and 30% sheltered homeless persons, and the final sample consisted of 68% unsheltered and 32% sheltered persons. The total sample was deemed sufficient given the study purpose was to explore associations between the key predictors and the outcome without any hypothesis testing. Individuals were eligible if they were at least 18 years old and had at least six months' experience of homelessness in Accra.

## Procedure

Approvals were obtained from the Western Sydney University Human Research Ethics Committee, the Ghanaian Department of Social Development, and four community organisations

that serve people experiencing homelessness in Accra. Respondents were recruited by trained research assistants experienced in public health research with vulnerable populations and fluent in local languages. For potential respondents in the streets and slums, their community leaders were contacted in advance to facilitate data collection. Respondents were recruited day and night during pre-existing outreach, health screening, and free meal programmes. The research team was introduced to the potential respondents, research information and consent were discussed, and consenting individuals were invited to pre-arranged community centres or playing grounds for participation. At the shelters, staff members introduced the research team to clients for information dissemination, consent, and participation.

To prevent exploitation, individuals with serious cognitive impairment, intellectual disability, and mental illness were excluded if the conditions limited capacity to consent. For such individuals, the capacity to provide consent was determined by social workers and shelter staff based on the seriousness of the health condition and prior experience of serious distress. Intoxicated individuals were only engaged when they had the ability to give informed consent. Three people judged as incapable to provide consent were excluded. To prevent coercion, community leaders, social workers, and shelter staff members were not involved in the consent process except when a person's capacity to consent was questionable. Respondents had the opportunity to confirm or re-negotiate consent over time. That is participants who had already agreed to participate were asked of permission again prior to the start of the survey. Participants could change their decision to participate or withdraw their participation altogether based on changes in personal lives or availability of new information about the research.

Activities were conducted in either the local Ghanaian language (66%, n = 201) or English (34%, n = 104). The research assistants read the questions to respondents, recorded answers in electronic format, and then synchronised to the online software platform CANVAS (https://canvass.acspri.org.au) using android tablets. The survey lasted for 50 to 60 minutes, and respondents were reimbursed with mosquito repellent (valued at AU$5). All respondents could also check-in to any of the two partner alcohol and drug rehabilitation centres with the help of social workers free of charge. Community organisations provided timely referrals for mental health support for respondents who may experience psychological harm and distress. Two respondents experienced mild discomfort answering questions on discrimination and violence which led to a temporary pause of the survey.

## Variables

**Outcome variable: Alcohol and drug consumption.** Substance use risk levels were assessed using the ASSIST version 3.0, a tool to identify levels of risky substance use and related health risks [35]. The ASSIST consists of eight questions on lifetime and past three-month use of nine substances: tobacco, alcohol, cannabis, cocaine, amphetamine-type stimulants, inhalants, sedatives, hallucinogens, and opioids. For each substance used in the last three months, further questions assess the following: 1) frequency of use, 2) urges to use, 3) health, social, legal, or financial problems, 4) and interference with role responsibilities. The ASSIST also queries failed attempts to reduce substance use and whether someone ever expressed concern about use; a follow-up question determines if this occurred in the past three months or more than three months ago [7, 35].

A Specific Substance Involvement (SSI) score was calculated for each substance (range: 0–39, same for total scale) using standard ASSIST scoring procedures. Individual SSI scores were categorised into three risk levels: low, moderate, high (Table 1) [13]. An overall substance use risk variable was defined based on the highest risk level assigned for any substance.

**Table 1. ASSIST-defined substance use risk categories- specific substance involvement score (SSI).**

| Substance | Score | Risk Level/ Outcome |
|---|---|---|
| Alcoholic Beverages | 0–10 | Low |
| | 11–26 | Moderate |
| | ≥ 27 | High |
| Other Substances (Tobacco Products, Cannabis, Cocaine, Amphetamine type stimulants, Inhalants, Sedatives or Sleeping Pills, Hallucinogens, Opioids) | 0–3 | Low |
| | 4–26 | Moderate |
| | ≥ 27 | High |

The WHO-ASSIST has a test–retest reliability coefficient as high as 0.90 [36]. Although not used in any prior study in Ghana, the instrument has been used and validated with the sub-Saharan African countries like Nigeria, reporting internal correlation coefficients of greater than 0.7 [37].

**Explanatory variables.** The following explanatory variables were selected based on previously published studies among persons experiencing homelessness in both developing and developed countries [5, 7, 38, 39]:

1. *Sociodemographic characteristics*: Age, gender, education level, intimate relationship, employment status, and income levels.

2. *Migration and homelessness characteristics*: Accra migration status (migrant/non-migrant). Homelessness characteristics included shelter status (sheltered/unsheltered). Sheltered homelessness refers to temporarily residing in a homeless shelter, crisis and emergency accommodations, and residential drug and alcohol rehabilitation facilities. Unsheltered homelessness, on the other hand, refers to roofless condition, such as the streets, market-places, alleys, abandoned vehicles, under bridges, verandas, and other places not designed for shelter. Total lifetime duration of homelessness was also measured (chronic- ≥1 year, non-chronic- <1 year).

3. *Experiences of stigmatisation*: Assessed using a 12-item Social Stigma Scale to measure the experience of general stigma (10 items; scores ranged from 10–40) and self-blame (2 items; scores ranged from 2 to 8) [40]. Items are scored on the following 4-point Likert-type scale: strongly agree = 4, agree = 3, disagree = 2 and strongly disagree = 1. Higher scores imply higher levels of stigma and self-blame experienced. The Social Stigma Scale has an overall internal reliability of 0.87, with the subscales of self-blame and general stigma having Cronbach's reliability coefficients of 0.78 and 0.89 respectively [40]. The scale has been used and validated with the homeless population in Ghana [41].

4. *Health characteristics*: Self-reported experience or diagnosis of a range of physical and mental health conditions in the past three months. Presence of physical health problem (yes/no) or mental health problem (yes/no) was defined as having a positive response to at least one condition.

5. *Violence and abuse*: measured using items from the Ghana Living Standard Survey (GLSS) Round 6 [15]. Three types of violence within the last six months were assessed in line with previously published studies among homeless and other impoverished people [42, 43]:

a. Physical violence—defined as being hit, slapped, kicked, bitten, choked, twisted arm, kidnapped, struck with an object, pulled hair, spat on, urinated on, or the victim of a robbery with a weapon.

b. Emotional violence—defined as any experience of threats, harassment, intimidation, controlling behaviour, insults, items seized, cruelty.

c. Sexual violence—defined as holding, grabbing, and touching certain parts of a person in a manner that irritates or angers the person involved, and being forced to have sex of any kind.

To limit the influence of language barrier and allow participation by those who only spoke the local language, the survey questionnaire, including the ASSIST and the Social Stigma Scale was constructed in English, translated into Akan, and back-translated into English to ensure the actual meaning of items was preserved in the translation. As part of the back-translation, a list of culture-specific names of the diseases, common substances of abuse, and key terms were provided side-by-side their English names.

## Statistical methods

Frequencies, percentages and medians were used to describe respondents' characteristics and substance use risk levels (i.e. low, moderate, high). Logistic regression modelling was used to examine significant factors associated with high-risk substance use (ASSIST score $\geq$27). Factors included in the analyses were the socio-demographic characteristics, Accra migration and homelessness variables, social stigma, and self-reported physical and mental health problems. Two measures of violence (i.e. physical/emotional and sexual) were also included. Physical and emotional violence were merged into one measure (i.e. experience of physical or emotional violence, dichotomised as Yes = 1 and No = 0) due to high correlation (***r = .96)***.

Separate multivariable logistic regression analyses were conducted for high-risk use of alcohol, cocaine, and cannabis due to their particularly high prevalence in both the general Ghanaian population and this study's sample. Variable selection techniques consisted of two separate logistic regression analyses as follows:

1. For each measure of high-risk substance use, a bivariate logistic regression was conducted separately for the explanatory variables.

2. All variables with p-value $\leq$.25 in the bivariate analysis [44, 45] were included in the multivariable logistic regression. Variables were excluded as follows:

a. Age, education, relationship status, and migration status were excluded from analysis of overall high-risk substance use.

b. Age, education. relationship status, income, and shelter status were excluded from the analysis of high-risk alcohol use.

c. Education, income, migration status, and shelter status were excluded from the analysis of high-risk alcohol use.

d. Relationship status was excluded from the analysis of high-risk cocaine use.

However, self-blame (a sub-category of social stigma) and presence of mental health problem were included apriori in all analysis. International literature shows that that people engage in problematic substance use as a coping strategy for stress, trauma, and stigma from violence [46, 47]. Similarly, there is an established relationship with substance use in the homeless population [7].

Adjusted Odds Ratios (AORs) with 95% confidence intervals (CIs) were estimated as the measure of association between the explanatory and outcome variables. All statistical analyses were conducted in Stata version 14.0 (Stata Corp, College Station, TX, USA). All variables with a p-value≤.05 were considered significant. Multicollinearity was checked after the regressions. Interactions were examined between gender and various measures of violence, as well as between physical/emotional and sexual violence; none of them were statistically significant (S1 Table).

## Results

### Participants

The final sample size of 305 consisted of more men (61%) with a median age of 32 years (interquartile range [IQR] = 17 years; Table 3). Nearly three-quarters had, at best, a year 12 education; and most respondents (54%) were not in any intimate relationship. Those in relationships were legally or customarily married or in a consensual union. Most respondents identified themselves as employed (63%), commonly as street vendors, truck loaders or pushers, car washers, caterers, and cleaners. The median net monthly income was US$76.

Most respondents were urban migrants in the Accra city (78%), including non-Ghanaians. Two-thirds were unsheltered sleeping in streets, marketplaces, under bridges, and construction sites. Most respondents were considered chronically homeless (median homelessness duration = 5 years, IQR = 7, Table 3). More than three-quarters had experienced high levels of general stigma (median = 26, IQR = 9; range = 10–40) and self-blame (median = 5, IQR = 2; range = 2–8) due to homelessness. Self-reported physical and mental health problems were 72% and 87% respectively. Eye problems, blood pressure, dental problems, and respiratory problems were the top four physical health problems. Commonly reported mental health problems were tension and anxiety, depression, restlessness and gloominess, and suicidal thoughts. Emotional, physical, and sexual violence during the past six months were reported by 34%, 32%, and 19% of respondents respectively (Table 3).

### Substance use prevalence and risk levels

**Lifetime and recent substance use.** The lifetime prevalence of any substance use was 71% (n = 216) while 59% reported recent use (Fig 1). The substances most used were alcoholic

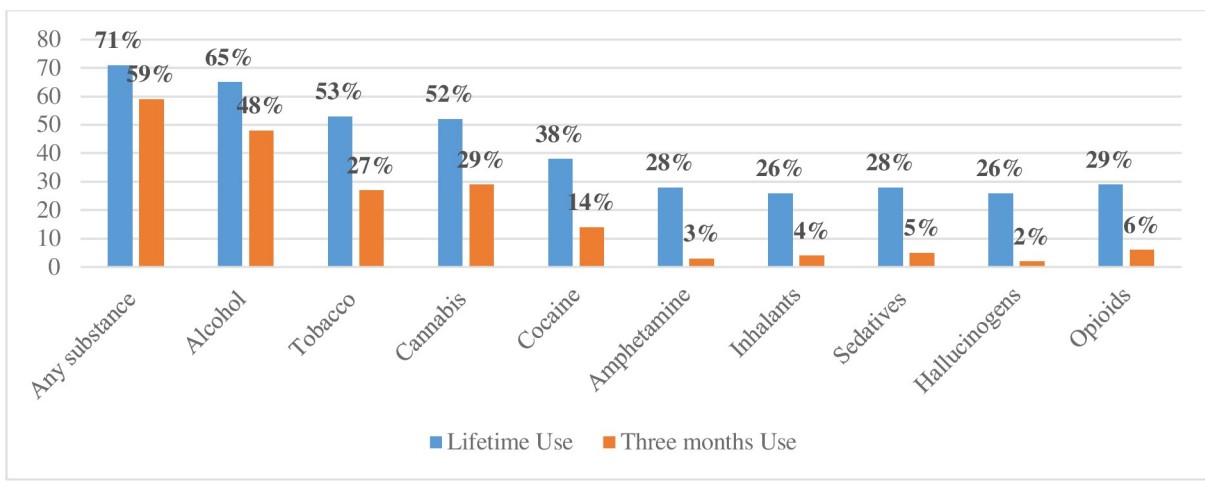

**Fig 1. Lifetime and recent substance use among people experiencing homelessness, stratified by substance type (n = 305).**

**Table 2. Substance use risk categories.**

|  | Highest Specific Substance Index (N = 216) | Alcohol (N = 197) | Tobacco (N = 160) | Cannabis (N = 158) | Cocaine (N = 115) | Opioids (N = 89) | Amphetamine (N = 85) | Sedatives (N = 85) | Inhalants (N = 79) | Hallucinogens (N = 78) |
|---|---|---|---|---|---|---|---|---|---|---|
|  | n (%) | n (%) | n (%) | n (%) | n (%) | n (%) | n (%) | n (%) | n (%) | n (%) |
| Low Risk | 11 (5.1) | 19 (9.6) | 52 (32.5) | 51 (32.3) | 54(47.0) | 61 (68.5) | 69 (81.2) | 67 (79.3) | 64 (81.0) | 65 (83.3) |
| Moderate Risk | 119 (55.1) | 126 (64.0) | 90 (56.3) | 85 (53.8) | 29 (25.2) | 21 (23.6) | 16 (18.8) | 18 (21.2) | 14 (17.7) | 10 (12.8) |
| High Risk | 86 (39.8) | 52 (26.4) | 18 (11.3) | 22 (13.9) | 32 (27.8) | 7 (7.9) | 0 | 0 | 1 (1.3) | 3 (3.9) |

Note: Highest Specific Substance Index has been reported as the overall risk

beverages (65% lifetime, 48% recent), followed by tobacco products (53% lifetime, 27% recent), cannabis (52% lifetime, 29% recent), and cocaine (38% lifetime, 14% recent). Consumption of other illicit substances was low.

**Risk related to substance use (WHO ASSIST-Defined).** Almost all the 216 respondents that had ever used substances were classified as engaging in risky substance use (95%) with 55% at moderate risk and 40% at high risk (Table 2). Moderate-risk use predominated for alcohol (64%), tobacco (56%) and cannabis (54%). For cocaine, high- and moderate-risk use were comparable (27% and 25%, respectively). For other illicit substances (hallucinogens, sedatives, inhalants, and opioids), most use was low-risk (69%-83%; Table 2).

**Respondents' demographic, migration and homelessness characteristics by highest specific substance involvement (SSI) score.** Bivariate analyses indicated substantial differences in respondents' substance use risk levels based on socio-demographic, migration, and homelessness characteristics (Table 3). Moderate/high-risk substance users were significantly more likely to be men than women (80% vs 48%, p<.001), unemployed than employed (76% vs 62%, p<.050), and chronically versus recently homeless (70% vs 51%, p<.010). Moderate/high-risk substance users were also more likely to have experienced general stigma than not (72% vs 51%, p<.001), and have a physical health problem than not (72% vs 55%, p<.010). A higher proportion of survivors of physical violence (87% yes vs 58% no, p<.001), emotional violence (85% yes vs 58% no, p<.001), and sexual violence (86% yes vs 63% no, p<.001) had moderate/high-risk substance use. On the other hand, low-risk users or lifetime abstainers were significantly more likely to be women, homeless for less than a year, non-victims of violence, and had no experience of stigma. No significant differences were observed in substance use risk levels based on migration status, current shelter status, and self-reported mental health problems (Table 3).

**Factors associated with a high-risk substance use.** Engaging in high-risk substance use was significantly associated with the experience of violence (Table 4). High-risk use of at least one substance was much more prevalent among survivors of sexual violence (AOR = 3.94; 95%CI 1.85–8.39, p<.001) and physical or emotional violence (AOR = 3.54; 95%CI 1.89–6.65, p<.001). In particular, physical or emotional violence survivors had higher odds of engaging in high-risk use of cocaine (AOR = 4.67; 95%CI 1.60–13.62; p = .005), alcohol (AOR = 3.42 95%CI 1.68–6.95, p<.001), and cannabis (AOR = 3.05; 95%CI 1.02–9.14; p = .050). Sexual violence survivors were nearly nine times more likely to have high-risk use of cocaine compared to non-victims (AOR = 8.82; 95%CI 2.83–27.52; p< .001).

Gender and income levels were also associated with substance use. The likelihood of engaging in high-risk use of at least one substance was significantly higher for men (AOR = 4.09; 95%CI 2.06–8.12, p<.001, Table 4), particularly for cocaine (AOR = 7.52; 95%CI 2.26–24.99, p = .001), cannabis (AOR = 4.23; 95%CI 1.23–14.58, p = .022), and alcohol (AOR = 2.75; 95%

**Table 3. Characteristics of study respondents by substance use risk levels.**

| Variable | All Participant (N = 305) | Moderate/High Risk Users (N = 205) | Low Risk/Abstainers (N = 100) | p-value |
|---|---|---|---|---|
| | n (%) | n (%) | n (%) | |
| Gender | | | | |
| Men | 185 (61) | 148 (80) | 37 (20) | < .001 |
| Women | 120 (39) | 57 (48) | 63 (52) | |
| Age Categories | | | | |
| 18–24 | 74 (24) | 44 (59) | 30 (41) | .215 |
| 25–34 | 91 (30) | 60 (66) | 31 (34) | |
| 35–44 | 70 (23) | 49 (70) | 21 (30) | |
| 45–54 | 45 (15) | 36 (80) | 9 (20) | |
| 55 or above | 25 (8) | 16 (64) | 9 (36) | |
| Education | | | | |
| Never been to school | 75 (24) | 47 (63) | 28 (37) | .588 |
| Basic | 154 (51) | 107 (69) | 47 (31) | |
| Secondary/Tertiary | 76 (25) | 51 (67) | 25 (33) | |
| Employment | | | | |
| Employed | 192 (63) | 119 (62) | 73 (38) | .011 |
| Unemployed | 113 (37) | 86 (76) | 27 (24) | |
| Income Category | | | | |
| Lower | 156 (52) | 111 (71) | 45 (29) | .122 |
| Middle-lower | 132 (44) | 81 (61) | 51 (39) | |
| Middle-upper | 11 (4) | 9 (82) | 2 (18) | |
| Relationship status | | | | |
| No relationship | 165 (54) | 117 (71) | 48 (29) | .159 |
| In a relationship | 139 (46) | 88 (63) | 51 (37) | |
| Migration status | | | | |
| Migrant | 237 (77.7) | 154 (65) | 83 (35) | .121 |
| Non-migrant | 68 (22.3) | 51 (75) | 17 (25) | |
| Current Shelter Status | | | | |
| Unsheltered | 206 (67.5) | 144 (70) | 62 (30) | .149 |
| Sheltered | 99 (32.5) | 61 (62) | 38 (38) | |
| Length of Homelessness | | | | |
| < 1 year | 51 (17) | 26 (51) | 25 (49) | .007 |
| ≥ 1 year | 254 (83) | 179 (70) | 75 (30) | |
| Reported Self-blame Experience | | | | |
| Yes | 182 (60) | 126 (69) | 56 (31) | .360 |
| No | 123 (40) | 79 (64) | 44 (36) | |
| Reported General Stigma | | | | |
| Yes | 240 (79) | 172 (72) | 68 (28) | .001 |
| No | 65 (21) | 33 (51) | 32 (49) | |
| Physical Health Problem (s) | | | | |
| Present | 220 (72) | 158 (72) | 62 (28) | .006 |
| Not present | 85 (28) | 47 (55) | 38 (45) | |
| Mental Health Problems | | | | |
| Present | 264 (87) | 180 (68) | 84 (32) | .360 |
| Not present | 41 (23) | 25 (61) | 16 (39) | |
| Reported Physical Violence | | | | |

(*Continued*)

**Table 3.** (Continued)

| Variable | All Participant (N = 305) | Moderate/High Risk Users (N = 205) | Low Risk/Abstainers (N = 100) | p-value |
|---|---|---|---|---|
| | n (%) | n (%) | n (%) | |
| Yes | 100 (33) | 87 (87) | 13 (13) | < .001 |
| No | 205 (67) | 118 (58) | 87 (42) | |
| Reported Emotional violence | | | | |
| Yes | 103 (34) | 88 (85) | 15 (15) | < .001 |
| No | 202 (66) | 117 (58) | 85 (42) | |
| Reported Sexual Violence | | | | |
| Yes | 59 (19) | 51 (86) | 8 (14) | < .001 |
| No | 246 (81) | 154 (63) | 92 (37) | |

CI 1.30–5.84, p = .008). Individuals in the middle-income group had lower odds of high-risk use of at least one substance (AOR = .42; 95%CI 0.23–0.78, p = .006), particularly for cocaine (AOR = 0.21; 95%CI 0.07–0.62, p = .004). Stigmatisation and self-reported health problems did not have significant associations with high-risk substance use. Separate logistic regression analyses were attempted for sheltered and unsheltered respondents but were unsuccessful due to small sample sizes after separation.

## Discussion

This study set out to explore risky substance use and associated factors in adults experiencing homelessness in Ghana's capital, Accra. Substance use was highly prevalent and most use was at the level of moderate to high risk of harm. Experiences of violence, male gender and low-income status were associated with more risky substance use. Contrary to expectations, experiences of stigma and mental health problems were not statistically related to risky substance use.

The high prevalence of risky substance use in the sample may be expected given the easy access to alcohol and drugs in Ghanaian cities [48, 49]. Since the early 2000s Ghana has been an important international trafficking route for cocaine, cannabis, and heroin from Asia and South America to other African countries, United States, and Europe [50–52]. Alcohol and cannabis are locally produced and trafficked from Ghana to other African countries [53], and methamphetamine has been locally produced and transported to East Asia since 2009 [17, 53]. The production and/or trafficking of these substances has contributed to the establishment of a ready local market and a surge in domestic consumption [17], including consumption among vulnerable populations. Particularly, people experiencing homelessness in Ghanaian cities are mostly exposed to places of high drug activity with easy accessibility [17]. Low-income status may play a crucial role in people's access and use of substances through engagement in sale and distribution to generate income [54].

It is noteworthy that close to a third of this study's sample had never used any substance in their lifetime and 5% of all users were at low risk of harm despite their generally low-income status and exposure to high drug activity. Resilience factors such as self-esteem [55–57], intellectual ability [58–61], religiosity/spirituality [55, 62], and personal skills [55, 63, 64] may account for the non-use. Individual's attitudes about substance use including fear of consequences (health, social, legal), personal and family norms, and lack of interest may also explain their non-use [65–67]. Future research should focus on non-engagement in substance use and associated factors among people who experience homelessness. Knowing the extent and reasons/factors associated with non-engagement in substance use or misuse could inform policies

**Table 4. Adjusted odds ratios for having assist-defined high-risk substance use.**

| VARIABLES | All substances AOR (95%CI) | P-value | Alcohol AOR (95%CI) | P-value | Cocaine AOR (95%CI) | P-value | Cannabis AOR (95%CI) | P-value |
|---|---|---|---|---|---|---|---|---|
| **Gender** | | | | | | | | |
| Women | Reference | | Reference | | Reference | | Reference | |
| Men | 4.09 (2.06–8.12) | < .001 | 2.75 (1.30–5.84) | .008 | 7.52 (2.26–24.99) | .001 | 4.23 (1.23–14.58) | .022 |
| **Age** | - | | - | | 0.98 (0.95–1.02) | .354 | 0.96 (0.92–1.00) | .076 |
| **Education Level** | | | | | | | | |
| No formal education | | | | | Reference | | | |
| Basic Education | - | - | - | - | 1.15 (0.41–3.19) | .790 | - | - |
| Secondary/Tertiary | | | | | 0.42 (0.10–1.81) | .248 | | |
| **Relationship Status** | | | | | | | | |
| No Relationship | - | - | Reference | | - | - | Reference | |
| In a relationship | | | 0.89 (0.45–1.76) | .747 | | | 0.52 (0.18–1.45) | .208 |
| **Income Level** | | | | | | | | |
| Low income | Reference | | | | Reference | | | |
| Middle income | 0.42 (0.23–0.78) | .006 | - | - | 0.21 (0.07–0.62) | .004 | - | - |
| **Migrant status** | | | | | | | | |
| Non-Migrant | - | - | - | - | Reference | | - | - |
| Migrant | | | | | 0.40 (0.11–1.48) | .170 | | |
| **Shelter Status** | | | | | | | | |
| Sheltered | Reference | | | | Reference | | | |
| Unsheltered | 1.17 (0.60–2.29) | .645 | - | - | 0.85 (0.28–2.60) | .775 | - | - |
| Self-Blame | 1.06 (0.87–1.30) | .533 | 1.24 (0.99–1.55) | .059 | 0.97 (0.71–1.32) | .847 | 0.90 (0.65–1.27) | .560 |
| General Stigma | 1.02 (0.96–1.08) | .592 | 0.99 (0.93–1.06) | .857 | 1.02 (0.93–1.13) | .623 | 0.99 (0.89–1.09) | .805 |
| **Experienced Physical/Emotional violence** | | | | | | | | |
| No | Reference | | Reference | | | | | |
| Yes | 3.54 (1.89–6.65) | < .001 | 3.42 (1.68–6.95) | .001 | 4.67 (1.60–13.62) | .005 | 3.05 (1.02–9.14) | .046 |
| **Experienced Sexual violence** | | | | | | | | |
| No | Reference | | Reference | | | | Reference | |
| Yes | 3.94 (1.85–8.39) | < .001 | 1.41 (0.64–3.12) | .399 | 8.82 (2.83–27.52) | < .001 | 2.43 (0.80–7.37) | .115 |
| **Self-Reported Physical Illness** | | | | | | | | |
| No | Reference | | Reference | | Reference | | Reference | |
| Yes | 1.52 (0.74–3.12) | .257 | 2.24 (0.91–5.53) | .080 | 1.27 (0.40–4.03) | .685 | 1.07 (0.35–3.29) | .911 |
| **Self-Reported Mental Illness** | | | | | | | | |
| No | Reference | | Reference | | Reference | | Reference | |
| Yes | 1.17 (0.45–3.07) | .746 | 1.23 (0.38–3.95) | .728 | 0.89 (0.17–4.58) | .886 | 0.47 (0.13–1.69) | .247 |
| Number of Observations | 299 | | 304 | | 299 | | 304 | |

Note: Empty spaces represent where variables were excluded based on the p<0.25 cut-off.

and interventions based on strength and resilience factors among people experiencing homelessness and other vulnerable populations.

The high prevalence and risky substance use in this Ghanaian study broadly agree with previous findings in Africa [68–71] and Western countries [4–7, 72–75]. Although generally high, differences in prevalence exist between the present study and previous research which may be explained by the differences in defining homelessness as well as substance use assessment tools. The present study included a broad homeless population consisting of unsheltered and sheltered people such as those undergoing rehabilitation. A previous study in Ghana and

South Africa that used a narrow definition of unsheltered homelessness reported over 80% life-time substance use [71]. Conversely, an American study [7] of people using homeless shelters and primary health care settings (where high drinking and drug use may be prohibited) reported a substantially lower prevalence. These differences in prevalence suggests that substance use may be higher among those experiencing rooflessness compared to other forms of homelessness. Unfortunately, comparisons between sheltered and unsheltered respondents in the present study could not be conducted due to the small sample sizes of the two subgroups. In terms of assessment tools, a Canadian study [5] that assessed problematic alcohol drinking and drug use using Alcohol Use Disorder Identification Test (AUDIT) and the Drug Abuse Screening Test (DAST) respectively reported substantially lower prevalence for alcohol (16%) but a similar prevalence for drugs (29%). However, it is noteworthy that the American study [7] that used ASSIST reported a substantially lower prevalence of problematic alcohol and drug use.

The finding that a high proportion of individuals engaged in risky substance use implies the experience of moderate to severe health problems and suggests the need for brief interventions and more intensive treatments [12]. More critical is the high proportion of people engaged in the risky use of more than one substance which suggests multiple and complex psychosocial risks requiring integrated treatment [17, 66, 76]. While abstinence-based interventions through rehabilitation are important [17, 77], outpatient rehabilitation detoxification services (which are mostly available in Accra) may be unsuitable for people experiencing homelessness. Ghanaian homeless individuals have to return to places of high drug activity, peer influence, violence, and other traumatic conditions that may cause relapse and treatment dropout. Residential rehabilitation may therefore be more suitable for people experiencing homelessness particularly for survivors of violence and youth under gang pressure [78, 79]. Unfortunately, the number of residential rehabilitation facilities in Accra is woefully inadequate [17]. A small number of non-governmental organisations and religious bodies have established small-scale residential rehabilitation programmes targeting the homeless population, but the need for consent from a relative and separation from children prevents migrants and women with children from accessing services. Relaxing these requirements would help urban migrants and women with children–who constitute the majority of the homeless population in Accra–to access services. Equally critical is the inadequate aftercare programmes such as rehousing and income generating activities for residential rehabilitation patients upon discharge. Absence of such programmes implies a likely return to homelessness and subsequent relapse. Linking supportive housing services [80, 81] and skills training [82, 83] to substance use treatments are recommended. A few faith-based rehabilitation programmes in Accra offer clients opportunities to earn a decent income upon discharge through vocational training, education, sports, and peer-mentorship, while some clients are assisted to reunite with their families. Government investment in such initiatives would help increase the capacities of rehabilitation centres to admit more clients experiencing homelessness, and to help successful clients transition into stable housing through income-earning activities.

Also important are targeted harm-reduction interventions that assist people to identify and address specific factors behind their substance use, reduce usage rates, and stop harmful practices such as needle sharing even if they are not motivated to abstain [13, 17, 84]. Embedding such approaches into the government and donor agencies' strategies is important particularly for the homeless population who may engage in risky substance use to cope with the poor environment, experiences of stigmatisation and isolation, and stress [42, 85–90]. Prior studies have recommended harm-reduction interventions through counselling, education, and peer support on self-help strategies for reducing substance use and other harmful practices [91]. The community organisations and rehabilitation centres in Accra and other Ghanaian cities

could expand their periodic outreach, health screening, and meal programmes in the streets and slums to accommodate such strategies.

High levels of violence found in this study are consistent with other studies of homeless populations in Africa [68, 70, 92, 93] and Western countries [42, 94–97]. The multivariable analysis indicated that recent violence among people experiencing homelessness was related to risky substance use, as has been reported in Africa [69, 70], United States [42, 43, 90, 98] and Europe [86]. The association between risky substance use and violence may reflect the systemic violence inherent in the criminal networks involved in drug distribution, including enforcing debt payment and settling disputes between dealers and users, and among dealers (who are also mostly users) [99–102].

Along with the finding of the male gender associated with high-risk substance use, which confirms prior findings [7, 72, 103], the association between substance use and violence may also be attributed to substance use as a coping strategy for violence. This strategy includes using drugs to establish street reputation as 'hard core' and to gain entry into street gangs for protection against violence [73, 104]; using depressants to self-medicate injuries and ongoing pain from violence [85, 87, 89, 90, 105]; and consuming alcohol, cocaine, and opioids to appear tough and fearless and to prepare for fights [42, 85–87, 90]. Due to sample size issues, this study was unable to examine gender differences in relation to violence and substance use, but previous research have reported women's problematic substance use as a coping strategy for stress and trauma from violence [46, 47]. Meanwhile, women's lower odds of engaging in high-risk substance use in this study could be attributed to substantial under-reporting due to the greater social stigma for women's substance use. Women experience a high burden of criticism and widespread condemnation for engaging in substance use in many Ghanaian and African cultures [17, 106] which is often spoken about using the language of religion and perception of ungodliness [17]. For example, a qualitative study in Kenya reports that women who inject drugs experience gender-based stigma in societies and in healthcare settings because drug use ran contrary to gendered norms of behaviour [106]. One study demonstrated a higher prevalence of substance use among women compared to men [107], although the finding may have been confounded by younger age among women.

There were limitations in the conduct of this study. The sample's representativeness of the homeless population in Ghana could not be assessed due to the absence of data on the size and characteristics of the homeless population and the single-city sampling. Despite its non-random nature possible bias, and limited generalisability, the use of uncontrolled quota sampling allowed sampling from key sub-groups of the homeless population. Although accurate proportion of the various sub-groups was not possible due to unknown sizes, the uncontrolled quota sampling achieved some level of representativeness. The small sample from a single city also make also makes it difficult to make broad inferences about the findings from this study to the larger homeless population in Ghana. The study's use of self-reported data, including the assessment of risky substance use may introduce reporting bias and provision of socially desirable answers, given that most of the substances studied are illegal in Ghana. However, respondents were assured the researchers were under no legal obligation to report their engagement in substance use, and the respondents openly talked about their substance use experiences. Despite these limitations, this study makes a substantial contribution to the scarce research on substance use among adults experiencing homelessness in an African context. Adopting an inclusive definition of homelessness and assessment of substance use risk levels have expanded knowledge, as previous African studies only assessed the prevalence and types of substance use among street dwellers.

## Conclusions

This study demonstrates a high prevalence of risky substance use among adults experiencing homelessness in Accra, and the increased probability of engaging in high-risk substance use for survivors of violence, men, and low-income individuals. The findings highlight the need to provide targeted interventions to address risky substance use among the adult homeless population in the city. Interventions should take into consideration the complexity of what it means to be homeless in Ghana and Africa as a whole.

## Supporting information

**S1 Table. Logistic regression with interaction terms.**
(DOCX)

**S1 File.**
(DTA)

## Acknowledgments

The authors sincerely thank the Department of Social Welfare and Community Organisations working with people experiencing homelessness in Accra for approving the data collection. We also thank the social workers and community leaders in the streets and slums for their assistance in contacting potential participants. Special thanks to the respondents for their interest and participation.

## Author Contributions

**Conceptualization:** Benedict Osei Asibey, Brahmaputra Marjadi, Elizabeth Conroy.

**Data curation:** Benedict Osei Asibey, Brahmaputra Marjadi, Elizabeth Conroy.

**Formal analysis:** Benedict Osei Asibey, Brahmaputra Marjadi, Elizabeth Conroy.

**Investigation:** Benedict Osei Asibey, Brahmaputra Marjadi, Elizabeth Conroy.

**Methodology:** Benedict Osei Asibey, Brahmaputra Marjadi, Elizabeth Conroy.

**Project administration:** Benedict Osei Asibey, Brahmaputra Marjadi, Elizabeth Conroy.

**Software:** Benedict Osei Asibey.

**Supervision:** Brahmaputra Marjadi, Elizabeth Conroy.

**Validation:** Benedict Osei Asibey, Brahmaputra Marjadi, Elizabeth Conroy.

**Writing – original draft:** Benedict Osei Asibey, Elizabeth Conroy.

**Writing – review & editing:** Benedict Osei Asibey, Brahmaputra Marjadi, Elizabeth Conroy.

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
