## [Decision Letter · Decision Letter 0]

10 Jul 2022

PONE-D-21-35901Alcohol, Tobacco and drug use among adults experiencing homelessness in Accra, Ghana: A cross-sectional study of risk levels and associated factorsPLOS ONE

Dear Dr. Osei Asibey,

Thank you for submitting your manuscript to PLOS ONE. After careful consideration, we feel that it has merit but does not fully meet PLOS ONE’s publication criteria as it currently stands. Therefore, we invite you to submit a revised version of the manuscript that addresses the points raised during the review process.

We look forward to receiving your revised manuscript.

Kind regards,

Joel Msafiri Francis, MD, MS, PhD

Academic Editor

PLOS ONE

Journal Requirements:

Additional Editor Comments (if provided):

It would be more helpful to use the STROBE guidelines and format the manuscript accordingly including the use of appropriate heading, for example, study population rather than study respondent etc. 

Reviewers' comments:

Reviewer's Responses to Questions

**Comments to the Author**

1. Is the manuscript technically sound, and do the data support the conclusions?

Reviewer #1: Yes

Reviewer #2: Yes

2. Has the statistical analysis been performed appropriately and rigorously? 

Reviewer #1: Yes

Reviewer #2: Yes

3. Have the authors made all data underlying the findings in their manuscript fully available?

Reviewer #1: Yes

Reviewer #2: Yes

4. Is the manuscript presented in an intelligible fashion and written in standard English?

Reviewer #1: Yes

Reviewer #2: Yes

5. Review Comments to the Author

Reviewer #1: Abstract

Line 42: Please rephrase the sentence “The likelihood of engaging in high-risk substance used higher for men than women 43 (AOR=4.09; 95%CI 2.06-8.12, p<.001)”

-Did authors mean “the likelihood of engaging in high-risk substance use was higher for men than women?

Methods

-There are terms introduced in the text starting from the abstract that may require inclusion of their standard definition in the method section: What do the authors mean by “sheltered and unsheltered homelessness”

Line 112: “Since there was no official data on homelessness in Ghana, the uncontrolled quota sampling technique [34, 35] was used to obtain a representation of sheltered and unsheltered homeless populations”

-Please add justification for using uncontrolled quota sampling, how uncontrolled was the sampling procedure?

Line 157: Substance use risk levels were assessed using the ASSIST version 3.0, a tool to identify levels of risky substance use and related health risks

-For those who used local language, is there a translated validated ASSIST tool as per the local language? please specify, it will be great to also report the validity and reliability of the translated tool

Line 183: Experiences of stigmatisation: Assessed using a 12-item Social Stigma Scale to measure the experience of general stigma

-Same here, please report on the internal consistency and validity of the tool used

Line 192: Violence and abuse: measured using items from the Ghana Living Standard Survey

193 (GLSS) Round 6

- Same here, please report on the internal consistency and validity of the tool used

-Please specify the level of statistical significance used

Line 188: Self-reported experience or diagnosis of a range of physical and mental health conditions in the past three months

-Weren’t the patients who reported mental health conditions diagnosed in the past three months excluded? This was one of the exclusion criteria reported in the study participant section “individuals with cognitive impairment, intellectual disability, and mental illness were excluded from participation following advice from social workers and shelter staff.” Please clarify how comes those with mental health diagnosis were included.

Results

Line 241: Self-reported physical and mental health problems were 72% and 87% respectively

-Please explain how comes participants with mental health problems were included despite it being an exclusion criterion

Table 2: Please specify, if possible, in bracket that the “Highest Specific Substance Index” has been reported as the overall risk to make it more clear

Line 270: “Moderate/high-risk substance users were also more likely have experienced general stigma than not (271 72% vs 51%, p<.001)”

-add “to” have experienced

Discussion/Limitation

Line 342: “Although generally high, differences in prevalence exist between the present study and previous research”

-While comparing the magnitude of risky substance use in literature, authors should take note of the differences in the substance use assessment tools used in other studies and how they can be comparable/correlated with ASSIST tool.

Authors should highlight the disadvantages/ limitations that result from using uncontrolled quota sampling, biases resulting from non-randomization among others and how these were dealt with in the study

With interview-based questionnaire, there is a risk of information bias, provision of socially desirable answers, especially since the use of some of the drugs is illegal in most African countries. These too are usually anticipated in studies relating to substance use hence should be reported if relevant.

Discussion/Conclusion

Please also comment on generalizability of the study findings

Reviewer #2: 6 May 2022

PONE-D-21-35901

Alcohol, Tobacco and drug use among adults experiencing homelessness in Accra, Ghana: A cross-sectional study of risk levels and associated factors

Benedict Osei Asibey

Dear Authors,

Some consideration for your manuscript.

Line 42: missing words… substance use were higher

Line 46: Missing word… with the experience

Line 86: Could some additional information iro of this larger cross sectional study be given and where this piece of work fits in or even provide a reference to a paper possibly published on the larger study.

Line 144: What would re-negotiate consent entail, please just expand on this and under what circumstance such a situation is likely to occur.

Line 146: Did the fieldworker pose the question to the participant and enter it on behalf of the participant or were the questions posed and the participant themselves entered the response on the tablet. Possibility of bias which should be detailed in the limitations if not there.

Line 149: The incentive with repellant is an interesting one presumably because of high prevalence of malaria. If this could be stated.

Line 151: Were referrals to centres facilitated by the field workers? Were the participants all given the option to do this or was it based on a score on the ASSIST?

Line 238: Perhaps added into the methods section under the relevant sub-heading, the authors should indicate what baseline was used to determine chronic homelessness versus “non-chronic” homelessness.

Line 244: Presumably data on whether participants who self-reported an ailment sought help for physical/mental health problems was not collected? If yes it would be interesting to know.

Line 146: were women more like than men to have encountered sexual violence in the past 6 months. Although I later on pick up, this was not possible to determine due to sample size.

Line 271: more likely to have experience… missing word

Lines 259-363 – Is there are reference for this, because there may be a divergent view iro of the value/effectiveness on measured outcomes of outpatient treatment vs inpatient – there is evidence of the value of outpatient/community based services to. Perhaps the authors should do a quick review to look into this and provide references. This would apply to line 364 to mid-paragraph.

6. PLOS authors have the option to publish the peer review history of their article (what does this mean?). If published, this will include your full peer review and any attached files.

Reviewer #1: No

Reviewer #2: No

---

## [Decision Letter · Decision Letter 1]

2 Oct 2022

PONE-D-21-35901R1Alcohol, Tobacco and drug use among adults experiencing homelessness in Accra, Ghana: A cross-sectional study of risk levels and associated factorsPLOS ONE

Dear Dr. Osei Asibey,

Thank you for submitting your manuscript to PLOS ONE. After careful consideration, we feel that it has merit but does not fully meet PLOS ONE’s publication criteria as it currently stands. Therefore, we invite you to submit a revised version of the manuscript that addresses the points raised during the review process.

We look forward to receiving your revised manuscript.

Kind regards,

Joel Msafiri Francis, MD, MS, PhD

Academic Editor

PLOS ONE

Journal Requirements:

Reviewers' comments:

Reviewer's Responses to Questions

**Comments to the Author**

1. If the authors have adequately addressed your comments raised in a previous round of review and you feel that this manuscript is now acceptable for publication, you may indicate that here to bypass the “Comments to the Author” section, enter your conflict of interest statement in the “Confidential to Editor” section, and submit your "Accept" recommendation.

Reviewer #1: (No Response)

Reviewer #3: All comments have been addressed

2. Is the manuscript technically sound, and do the data support the conclusions?

Reviewer #1: Yes

Reviewer #3: Yes

3. Has the statistical analysis been performed appropriately and rigorously? 

Reviewer #1: Yes

Reviewer #3: Yes

4. Have the authors made all data underlying the findings in their manuscript fully available?

Reviewer #1: Yes

Reviewer #3: Yes

5. Is the manuscript presented in an intelligible fashion and written in standard English?

Reviewer #1: Yes

Reviewer #3: Yes

6. Review Comments to the Author

Reviewer #1: Line 131-133: Refer to “The initial plan was to sample 70% unsheltered and 30% sheltered homeless persons, and the final sample consisted of 68% unsheltered and 32% sheltered persons.”

Please move this statement that explains on the quota weightage just after mentioning quota sampling method.

Line 125-127: The authors report the sample size to be deemed sufficient to determine predictors. If possible, I would suggest reporting the power of the study at 5% significance level to support this.

Line 157: Substance use risk levels were assessed using the ASSIST version 3.0, a tool to identify levels of risky substance use and related health risks –

Line 173: This is a follow up from the previous comment: For those who used local language, is there a translated validated ASSIST tool as per the local language in Ghana? it will be great to also report the validity and reliability of the translated tool, same applies to the GLSS

NB: Some tools have been pre-translated into different languages for use by different localities, some of which will have studies reporting the validity of such translated tools. Hence, this question asks about the translated ASSIST tool, not the language used for the questionnaire – refer to the revision made by the authors on the Stigma tool

Line 298: Authors should be consistent with the statistical method used: is it univariate or bivariate? Please be consistent throughout

Line 320: Before reporting the AOR, authors may consider mentioning which variables were exclude in the multivariable model based on the p<0.25 cut-off as specified.

Line 340, table 4: Please clarify what the empty spaces in the table implies and if possible, indicate in the table legends. Indicate the long forms too in the legends for each table.

Line 340, table 4: Some factors that had p values >2.5 on bivariate analysis included in the multivariable model. For instance, the var Mental health problems and self-blame had p values of 0.36 on bivariate analysis, yet, they were included in the multivariable model. Please clarify.

NB: It is useful for the authors to mention the inclusion of factors that may be insignificant statistically but could be included in the model apriori, mental health problems is a good example here.

Line 381-384: “Although generally high, differences in prevalence exist between the present study and previous research”

-Mental health diagnosis varies as compared to other diseases, because of the presence of number of tools. While comparing the magnitude of risky substance use in literature, authors should take note of the differences in the substance use assessment tools used in other studies and how

they can be comparable/correlated with ASSIST tool (Please indicate where changes were made)

Line 453-454: The authors mentioned the representativeness of the quota sampling used for this study, please do also outline the limitation of using non-random sampling.

I would suggest a notable response to this, from previous comments: Please also comment on “generalizability of the study findings” ie. taking into account the sample size, are the authors able to make broad inferences about the findings from this study??

Reviewer #3: The current manuscript appears to be a revised version which has been resubmitted after fairly comprehensive revisions - this reviewer did not see the original version. As such, the paper seems fine. Any small issues are not worth another round of revisions.

7. PLOS authors have the option to publish the peer review history of their article (what does this mean?). If published, this will include your full peer review and any attached files.

Reviewer #1: **Yes: **Belinda J Njiro

Reviewer #3: **Yes: **Alasdair M Barr

---

## [Author Response · Author response to Decision Letter 1]

15 Nov 2022

The authors have attached a file containing our response to reviewers' comments

---

## [Decision Letter · Decision Letter 2]

29 Nov 2022

PONE-D-21-35901R2Alcohol, Tobacco and drug use among adults experiencing homelessness in Accra, Ghana: A cross-sectional study of risk levels and associated factorsPLOS ONE

Dear Dr. Osei Asibey,

Thank you for submitting your manuscript to PLOS ONE. After careful consideration, we feel that it has merit but does not fully meet PLOS ONE’s publication criteria as it currently stands. Therefore, we invite you to submit a revised version of the manuscript that addresses the points raised during the review process.

We look forward to receiving your revised manuscript.

Kind regards,

Joel Msafiri Francis, MD, MS, PhD

Academic Editor

PLOS ONE

Journal Requirements:

Additional Editor Comments (if provided):

Thanks for all the revisions. The paper reads well. Please kindly format the paper according to the journal format.

Reviewers' comments:

Reviewer's Responses to Questions

**Comments to the Author**

1. If the authors have adequately addressed your comments raised in a previous round of review and you feel that this manuscript is now acceptable for publication, you may indicate that here to bypass the “Comments to the Author” section, enter your conflict of interest statement in the “Confidential to Editor” section, and submit your "Accept" recommendation.

Reviewer #1: All comments have been addressed

Reviewer #3: All comments have been addressed

2. Is the manuscript technically sound, and do the data support the conclusions?

Reviewer #1: Yes

Reviewer #3: Yes

3. Has the statistical analysis been performed appropriately and rigorously? 

Reviewer #1: Yes

Reviewer #3: Yes

4. Have the authors made all data underlying the findings in their manuscript fully available?

Reviewer #1: Yes

Reviewer #3: Yes

5. Is the manuscript presented in an intelligible fashion and written in standard English?

Reviewer #1: Yes

Reviewer #3: Yes

6. Review Comments to the Author

Reviewer #1: (No Response)

Reviewer #3: (No Response)

7. PLOS authors have the option to publish the peer review history of their article (what does this mean?). If published, this will include your full peer review and any attached files.

Reviewer #1: **Yes: **Belinda Jackson Njiro

Reviewer #3: **Yes: **Alasdair M Barr

---

## [Author Response · Author response to Decision Letter 2]

1 Jan 2023

Response to editor's comments are attached

---

## [Editor Report · Decision Letter 3]

3 Jan 2023

PONE-D-21-35901R3Alcohol, Tobacco and drug use among adults experiencing homelessness in Accra, Ghana: A cross-sectional study of risk levels and associated factorsPLOS ONE

Dear Dr. Osei Asibey,

Thank you for submitting your manuscript to PLOS ONE. After careful consideration, we feel that it has merit but does not fully meet PLOS ONE’s publication criteria as it currently stands. Therefore, we invite you to submit a revised version of the manuscript that addresses the points raised during the review process.

Thanks for the revisions and the paper reads well. I have one critical comment - it would be helpful to standardise the reporting of  p values - decide whether to use 2 or 3 decimal points throughout the document.

We look forward to receiving your revised manuscript.

Kind regards,

Joel Msafiri Francis, MD, MS, PhD

Academic Editor

PLOS ONE

Journal Requirements:

Additional Editor Comments (if provided):

Thanks for the revisions and the paper reads well. I have one critical comment - it would be helpful to standardise the reporting of  p values - decide whether to use 2 or 3 decimal points throughout the document.
---

## [Editor Report · Decision Letter 4]

17 Jan 2023

Alcohol, Tobacco and drug use among adults experiencing homelessness in Accra, Ghana: A cross-sectional study of risk levels and associated factors

PONE-D-21-35901R4

Dear Dr. Osei Asibey,

We’re pleased to inform you that your manuscript has been judged scientifically suitable for publication and will be formally accepted for publication once it meets all outstanding technical requirements.

Kind regards,

Joel Msafiri Francis, MD, MS, PhD

Academic Editor

PLOS ONE
---

## [Editor Report · Acceptance letter]

20 Feb 2023

PONE-D-21-35901R4 

Alcohol, Tobacco and drug use among adults experiencing homelessness in Accra, Ghana: A cross-sectional study of risk levels and associated factors 

Dear Dr. Osei Asibey:

I'm pleased to inform you that your manuscript has been deemed suitable for publication in PLOS ONE. Congratulations! Your manuscript is now with our production department. 

Kind regards, 

on behalf of

Dr. Joel Msafiri Francis 

Academic Editor

PLOS ONE